# Detection Dogs Working in Hot Climates: The Influence on Thermoregulation and Fecal Consistency

**DOI:** 10.3390/ani14172456

**Published:** 2024-08-23

**Authors:** Leopold Slotta-Bachmayr, Antony Oyugi, Noreen Mutoro, Mary Burak, Mary Wykstra

**Affiliations:** 1Department Environment & Biodiversity, Salzburg University, 5020 Salzburg, Austria; 2Action for Cheetahs in Kenya, Nairobi P.O. Box 1611-00606, Kenya; oyugiantonyo@gmail.com (A.O.); mary.wykstra@actionforcheetahs.org (M.W.); 3School of the Environment, Yale University, New Haven, CT 06511, USA; mary.burak@yale.edu

**Keywords:** detection dog, body temperature, stool consistency, behavioral thermoregulation

## Abstract

**Simple Summary:**

The performance of search dogs is limited by their ability to cool down their body because the main cooling ability—panting—contradicts sniffing. We investigated the effect of hot environments on the general body condition of detection dogs working in Kenya. These dogs search for cheetah scats in their arid and hot habitat, where it is especially challenging for them to work. After a working day in the fields, these dogs showed a softer stool, which means a higher stress level, and elevated body temperature during the routine check-in the next morning. Our results showed that the use of search dogs in hot conditions is possible and useful but requires increased attention to prevent heat-related illness.

**Abstract:**

Body temperature is an important physiological parameter that influences the performance of working dogs. The main cooling mechanism in dogs is panting to support evaporative cooling, which reduces the dog’s ability to detect scents. In this study, we investigated the general body condition of four detection dogs searching for cheetah scats in a hot environment in northern Kenya. We evaluated the effect on the dog’s body temperature post-work in the short term (within hours) and long term (12–24 h). The fecal consistency and mean body temperature of the investigated dogs differed significantly between individuals but not between locations (moderate Nairobi and hot Samburu). On the morning after fieldwork, the dogs showed a significantly increased body temperature (37.9 ± 0.8 °C) compared to resting days (37.5 ± 2.2 °C). In the short term, on the first day of fieldwork, the dog’s body temperature (n = 2) decreased after 10 min of rest. On the second consecutive day of fieldwork, the 10-min recovery period was too short, and the body temperature did not decrease significantly. Our data showed that the use of detection dogs in hot conditions is possible and useful but requires increased attention to prevent heat-related illness.

## 1. Introduction

Body temperature is an important physiological parameter that influences the performance of working dogs and must be monitored to prevent life-threatening heat-related injuries, which may cause tissue cell damage or circulatory failure [1]. Several parameters that influence the dog’s body temperature include the dog’s physical characteristics (e.g., sex, mass, size, breed, coat colour, physical fitness), metabolic activity, and meteorological conditions such as ambient temperature, relative humidity, solar radiation, and wind speed. Taking all these factors into account, a Canine Thermal Model (CTM) has been developed to calculate how long military working dogs can safely work without overheating to maximize health and performance [2,3].

Generally, the rectal body temperature of dogs ranges between 37.5 °C and 39.0 °C, depending on the size and physical condition of the individual [4]. During physical activity such as treadmill exercises or sled dog races, the body temperature of dogs may increase to 43.3 °C since muscle activity generates heat [5,6,7]. Sniffing alone is documented to have a similar effect and has been observed in rescue dogs whose body temperature may reach up to 39.8 °C [8]. Mental stress may also contribute to increased body temperature [9].

Thermoregulation depends primarily on the ambient temperature, particularly during physical strain. With rising ambient temperature, many mammals (e.g., horses and humans) dissipate more heat by increased sweating and enhanced blood vessel perfusion in the skin [4], but dogs have sweat glands mainly on the soles of their paws [4]. The most important cooling mechanisms in dogs are through panting and increased perfusion of the skin vessels [4]. During panting, the skin on the tongue and in the muzzle is cooled off by water evaporation, which is improved by forced airflow resulting from increased inhalation and exhalation. Heat dissipation through panting is higher (approximately 60%) than conduction via the body surface [5,10]. During heavy panting, the dog’s ability to detect and identify different scents may decrease performance [11,12].

Besides physiological mechanisms, animals utilize behavioral thermoregulatory responses such as drinking water, choosing a cool resting spot, or decreasing locomotory activity (i.e., ‘slowing down’ [13]). Working dogs’ behavioral response during a search is controlled by the handler and usually consists of slowing down. Cooling down by water immersion, drinking water, or resting in a cool spot on the surface or in the shade is only possible when the handler allows the dogs to take a break.

In recent years, scat detection dogs have been increasingly used in wildlife management and conservation [14]. They are an excellent tool for locating scarce biological samples, such as scats, hidden in the vegetation and difficult for humans to find. Using scent as a biometric measure, dogs can identify and detect scats of different species with an accuracy of up to 100% [15]. Scat collection enables genetic and biochemical analysis to extract information on individuals, sex, relation to other populations, health, and hormonal status or diet [16,17,18].

Action for Cheetahs in Kenya (ACK) is a not-for-profit organization that provides science-driven solutions to conservation challenges. ACK utilizes four search dogs trained to locate wild cheetah scats and hair. Cheetahs live in arid and semi-arid hot habitats [19,20], where ACK deploys scat dogs. In addition to unfavorable hot temperatures during the day, safety and security concerns are also at night. To avoid safety concerns, especially hot ambient temperatures and direct solar radiation, the teams only work the dogs during the cooler morning or late afternoon hours.

The deployment of detection dogs in such a hot environment raises questions about the impact on the dog’s fecal consistency. In addition to being an animal welfare issue, this situation could also be problematic due to a decrease in the dog’s performance during a search. In this research study, we elaborate on the following research questions:What effect have training and search exercises had on the dog´s fecal consistency?How is the body temperature of scat dogs working in a hot environment affected on a short- (within hours) and long-term basis (12–24 h)?What strategies can be implemented for optimal deployment of working dogs in such environmental conditions?

## 2. Materials and Methods

### 2.1. Study Sites

The organization´s main base, which includes a facility to train dogs, is located near Nairobi, Kenya, in a predominantly agricultural area that is currently supporting residential and commercial activities. It lies at an average elevation of 1600 m above sea level, with an average annual precipitation of 745 mm and a mean temperature ranging from 12 °C to 28 °C [21,22].

Additionally, dogs are further trained and deployed to the field at a field station in Meibae Community Conservancy (MCC), Samburu, approximately 300 km north of Nairobi. This area lies in the greater Laikipia-Samburu-Isiolo ecosystem, a semi-arid region with low trees and bushes. This ecosystem hosts the second-largest resident cheetah population in Kenya [23]. MCC lies at an average altitude of 1100 m above sea level. The average precipitation is 210 mm, and the average temperature in most months is 30 °C [21,22].

In both the Nairobi and Samburu sites, the dogs are housed in kennels, which protect them from inclement weather or predators when they are not in training or conducting a field exercise.

### 2.2. Data Collection

This study is based on data collected from four neutered dogs (Table 1). All data was entered for each dog in their daily records between 1 April 2019 and 27 February 2022. Keeping conditions, feeding, training, and working was similar for all dogs except for the dog Warrior. Warrior was deployed several times to search for cheetah scats and had a heat stroke during a training session in May 2019. After the heat stroke, a heart murmur was diagnosed.

We collected data on each dog’s fecal consistency and body temperature. These variables were linked to various activities, which were broadly characterized as follows:Resting—days in which the dogs did not take part in training activity and only had routine physical condition checks, health breaks, grooming, and feeding (Nairobi, n = 448, Samburu, n = 459).Training—endurance sessions, nose work exercises approximately 1 to 2 h in the morning and the afternoon (Nairobi, n = 1457, Samburu, n = 1158).Field work—actual searches with the complete detection dog team in areas where recent cheetah sighting information was available (see [24]). Fieldwork was only carried out in Samburu and lasted between 30 min and 3 h. The dogs had less downtime when conducting surveys and were taken on more frequent short walks (Samburu, n = 36).Sick—when the dogs had an infection, were unwell, or were recovering from injury (Nairobi, n = 22, Samburu, n = 4).

The body condition score on a scale of 1–9 [25] and the health of each dog was evaluated daily in the morning through visual and tactile checks via a body check for injuries and parasites. During a morning walk, as a stress indicator, fecal scores [26,27] were assigned as well using the Purina Faecal Score (comp. [28]).

Dog body temperature was collected in two main ways: first, rectal temperature was collected during the daily morning checks using a standard digital fever thermometer (DTR-1221A, ±0.1 °C, iProven, Beaverton, OR, USA). Second, rectal body temperature was collected during exertion from endurance training, training searches, and/or field searches in order to understand how each dog’s body temperature fluctuated following exertion per session (n = 72). Temperature was taken at the start of the session before any exertion had occurred. However, if the session was scheduled within the next hour after the usual morning routine temperature check, then the initial daily temperature was also used as the exertion start temperature. The dog’s temperature was also taken at the end of the session and 10 min after the dog had rested in a shaded area. We also collected mean hourly temperature at 09:00 h from the nearest weather station (Wamba/Rift Valley; Weather Spark [22]) in order to assess the relationship between the change in body temperature and ambient temperature.

Notably, field searches with intensive documentation were carried out in Samburu in June and July 2019 before systematic documentation of the dog’s health started. Searches were conducted along a transect and documented using a GPS collar (position, length of the track, and search speed), and local weather data (temperature, humidity, wind direction, and wind speed) were recorded using a portable weather station [24]. The dogs’ search was stopped every 15 min, and dogs were provided with water and rested for at least 10 min. Additionally, during the search, the dogs were supported with a cooling vest, which reduces core body temperature [29]. The detailed data makes it possible to link behavioral and physiological data for the two older scat dogs (n = 22).

### 2.3. Analysis

We assessed the relationship between a dog’s morning body temperature and stool consistency with the previous day´s activity, individual dog, and location using a general linear model (GLM) analysis. This analysis was also performed for body temperature before and after exertion in relation to activity, individual, and location. Pearson correlation was used to test the relationship between behavioral and physiological data with environmental parameters or the duration of the exercise. All statistical analysis was done using SPSS 9.0.1 (SPSS, Chicago, IL, USA).

## 3. Results

### 3.1. Activity vs. Stool Consistency

The stool consistency showed a significant difference between the individuals (F = 3.534, *p* < 0.05) and a significant influence on the activity the day before (F = 5.454, *p* < 0.01). The location had no significant impact (F = 2.385, *p* > 0.05; Figure 1).

On average, the female Warrior had significantly softer stool (*p* < 0.05) compared to the other three dogs. The stool was significantly softer for all dogs after a day of fieldwork than after the other activities (*p* < 0.05). There was no influence of health status on stool consistency (Figure 1).

### 3.2. Variation in Body Temperature

The mean body temperature in the morning ranged between 35.0 °C and 39.8 °C depending on the individual, activity, and health status. Similar to stool consistency, the mean body temperature in the morning was significantly different between individuals (F = 6.736, *p* < 0.001) and was significantly influenced by the activity the day before (F = 8.189, *p* < 0.001). The location had no effect on body temperature (F = 0.742, *p* > 0.05).

To compare the body temperature of the different individuals, we only used data after resting days. After a resting day, the female Warrior (37.7 ± 0.3 °C) had a significantly higher body temperature than the other dogs (*p* < 0.05). Body temperatures were quite similar between the male Madi (37.4 ± 0.4 °C) and the remaining females Arti (37.5 ± 0.4 °C) and Persi (37.4 ± 0.4 °C) irrespective of the two different sites.

### 3.3. Activity vs. Body Temperature

The dog’s morning body temperature was significantly influenced by the previous day’s activity (F = 8.189, *p* < 0.001). In all four dogs, body temperature was not significantly different after a rest day (37.5 ± 2.2 °C) and a training day (37.5 ± 3.4 °C) but was significantly elevated on days following fieldwork (37.9 ± 0.8 °C) and when the dog was sick (37.8 ± 0.9 °C) (Figure 2).

### 3.4. Change of Body Temperature during Training and Field Work

Training sessions lasted from 20 min to 215 min (64 ± 42 min), and the duration of fieldwork was between 90 min and 235 min (189 ± 42 min). In the short term, body temperature changed significantly after exertion and 10 min after recovery. The mean body temperature at the start of exertion was 37.6 ± 1.3 °C and increased during exertion to 40.1 ± 0.9 °C. After 10 min of recovery, the body temperature decreased to 39.1 ± 0.6 °C. We found no influence of individual, location, type, and duration of activity on the three different times in which the body temperature was taken (Figure 3).

We also found a significant change in body temperature after exertion when the dogs were worked on consecutive days in the field. On the first day of fieldwork, we found a significant decrease in the dog’s body temperature from 39.6 °C to 38.7 °C after 10 min of rest (t = 4.214, *p* < 0.05). On the second consecutive day of fieldwork, the 10 min of recovery after exertion was too short, and the body temperature did not decrease significantly (t = 2.254, *p* > 0.05) from 39.8 °C to 39.0 °C (Figure 4). We did not find any influence of ambient temperature on the dog´s body temperature at the beginning, after exertion, and after 10 min of recovery.

### 3.5. Behavior vs. Ambient Temperature

We found that ambient temperature influenced the behavior of the dogs during fieldwork. There was a significant difference in search speed between the two dogs, Madi and Warrior, during fieldwork (Madi 7.1 ± 0.16 km/h, Warrior 6.4 ± 0.29 km/h, t = 6.974, *p* < 0.001), but there was no influence of consecutive days of fieldwork on search speed. For Madi, there was a significant negative relationship between search speed and ambient temperature (Figure 5, r = −0.615, *p* < 0.05), whereas for Warrior, no relationship between the two variables was detected.

## 4. Discussion

The high body temperatures of the scat dogs in this study result from long searches (up to 3 h) under hot conditions (ambient temperature 21–29 °C). Under moderate ambient conditions (around 10 °C) and exertions for one hour, the body temperature of rescue dogs can reach up to 42.5 °C [8]. In this case, body temperature is slightly influenced by sex and coat color [30]. We found no relationship between duration of exertion and body temperature. Data from the literature show that dogs heat up quite quickly, and the increase in body temperature may be slowed down because of good body condition, coat color, and coat structure [30]. After reaching a certain value, physiological mechanisms, mainly panting, regulate body temperature. This means that a relationship between body temperature and exercise duration can only be found at the beginning of an exercise, and additional regulation mechanisms prevent further increases in body temperature. However, panting interferes with smelling and sniffing, which, in turn, makes the dogs less effective in their tasks [31]. After strenuous physical activity (e.g., chasing a ball), dog performance declines by 10–15%, whereas heart rate and body temperature increase significantly [11,32]. When a dog’s duration of work increases, its physiological mechanisms for moderating body temperature must be supported by behavioral mechanisms. Specifically, for search dogs, dogs only have the ability to slow down, while the handler needs to be responsive to individual dogs and ensure that dogs are not reaching the stage of early physiological discomfort, offering some water and a place to rest in the shade. Studies have shown that search and rescue dogs regulate their heart rate by slowing down their pace [8]. Even after longer and repeated searches, the heart rate remains constant, but the dog´s activity decreases [8,32]. This may also be why the dogs in our study slowed down in hotter conditions: to reduce muscle activity, which is associated with heat production, and to help with thermoregulation of the body.

In the short term (e.g., within hours), our data show a significant increase in body temperature during physical activity across individuals, type and duration of activities, and ambient temperatures. This is likely due to the additive impact of physiological stress from physical activity, thereby resulting in a reduced thermoregulatory ability during the second day of fieldwork. Notably, fieldwork was the only activity that had a long-term effect (e.g., within 24 h) on the dog´s fecal consistency, thereby causing higher body temperature and softer stool the following day. During fieldwork, the dogs’ search was stopped every 15 min, and dogs were provided with water and rested for at least 10 min. Then, the search starts again and continues all in all for at least three hours. During this period, the detection dogs also showed behavioral thermoregulation by slowing down when searching in hotter conditions.

Whereas body temperature values observed in the morning after resting or training days are considered normal, the increase in body temperature in the short term, observed during training or fieldwork (up to 42.3 °C), is considered to be in the range of fever [33]. However, fever means the regulation of body temperature at an elevated level and clearly distinguishes between fever and elevations in body temperature that may result from passive heating. There is a large metabolic cost associated with elevating and then maintaining body temperature even 1 or 2 °C above normal [33]. Increased body temperature has been generally observed across different types of search dogs: Most commonly, dogs are known to reach body temperatures of up to 39.8 °C after short periods of searching [8]. Even for avalanche dogs, which work in much cooler environments, the mean body temperature after searching can surpass 39 °C [34]. On the other extreme, the body temperature of search and rescue dogs, which work under very hot conditions, can surpass 41 °C after 15 min of searching [35]. Lastly, after shorter searches (8.58 ± 2.49 min), the body temperature of a search dog may reach up to 40.6 °C [36]. Compared to these values, the body temperature data in our study showed similar values (40.1 ± 0.9 °C). Although we found no influence on ambient temperature, other studies have demonstrated that the body temperature of dogs working in cooler environments is typically lower than that of dogs working in hot conditions [8]. This is also true for body temperatures after strenuous activities like searching, agility, or chasing a ball [36].

If exertion continues and a dog’s ability to regulate heat production is exhausted, then a heat stroke may occur. Heat strokes are mainly related to a highly humid environment, excessive physical activity, lack of acclimation to the heat, and physical fitness [37,38]. The studied dogs did not develop heat stroke, but they showed increased body temperature even the following day (37.8 °C). O’Brien and Berglund [39] also found that insufficient breaks between periods of exertion lead to a constant increase in body temperature. This fact is also supported by the CTM. For the scat dogs in this study, body temperature on days after fieldwork is well below the critical core temperature of 41 °C, which requires medical attention [38]. Our data also show that the whole body of the dog is affected after days of fieldwork, resulting in softer stools as a sign of increased stress [26,27].

In the case of the dog Warrior, most of the data differ from that of the other dogs. This dog’s body temperature is significantly higher, and its feces are noticeably softer. Warrior’s searches were significantly slower than Madi’s, and there was no relation between search speed and ambient temperature. The data presented above may be related to the heat stroke the dog experienced after a training session and supported by the fact that dogs with a history of heat-related illness may predict altered heat sensitivity [40]. It may also indicate that the dog already had a heart condition that was not detected before and may have led to a heat stroke because of Warrior´s reduced thermoregulatory abilities.

One of the main limitations of the study may be the small sample size. However, some of the results are only anecdotal but very illustrative of the effects and consequences of dogs searching in hot environments. Moreover, even when two of the dogs were puppies when collection of data started, the small sample size is compensated by the long time series the data have been collected, so we can be optimistic that the obtained results, especially the statistically significant differences observed, are due to the environmental conditions. In any case, it is necessary to reflect on the difficulty of finding a larger number of dogs searching under these conditions and replicate the study under different climatic conditions to control the influence of the environment on the body temperature of a detection dog.

## 5. Conclusions

Our data demonstrates the influence of the exertion on search dogs working under hot environmental conditions. Body temperature is elevated not only because of the hot environment but also because of the long searches, which require physiological thermoregulation mechanisms to be supported by behavioral mechanisms. Under these conditions, a dog´s body is affected for a longer period, and this is evident from the elevated body temperature the next day and softer stool. This observation may help interpret whether the increase in body temperature is affected by the activity the day before or by a medical condition.

The duration of searches has to be shortened to 20–30 min to reduce the strain on the dogs, although even under this condition, the body temperature is elevated [11,35]. During a break, the dog has the ability to cool down, and the search should only continue after the dog stops panting. Besides increasing welfare, this also improves the performance of the dogs and results in a higher detection rate. If the search extends for a longer period and the body temperature remains elevated the next day, the dog should have the possibility to recover during this day. Our data have also shown that during the second day of fieldwork, the thermoregulatory abilities of the dogs are affected, which, again, led to a decreased performance. In conclusion, utilizing search dogs under hot conditions, such as those in northern Kenya, is possible and meaningful but is restricted by the dog´s ability of thermoregulation and requires increased attention to prevent heat-related illness and adapted search strategies to maintain high search performance.

## Figures and Tables

**Figure 1 animals-14-02456-f001:**
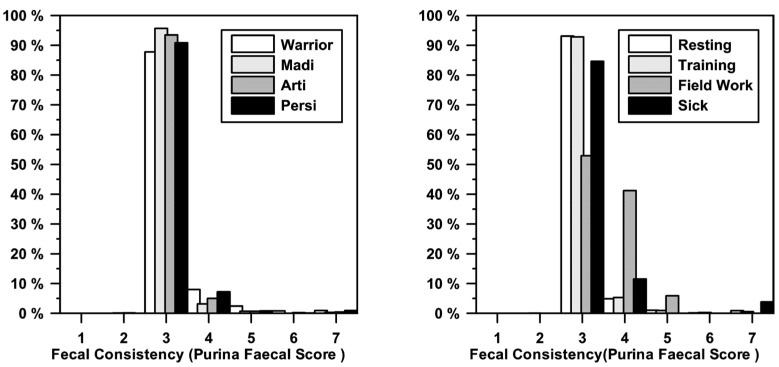
Change of fecal consistency in the morning in relation to different individuals and activities. There was a significant difference between the individuals (*p* < 0.05) and for different activities (*p* < 0.01).

**Figure 2 animals-14-02456-f002:**
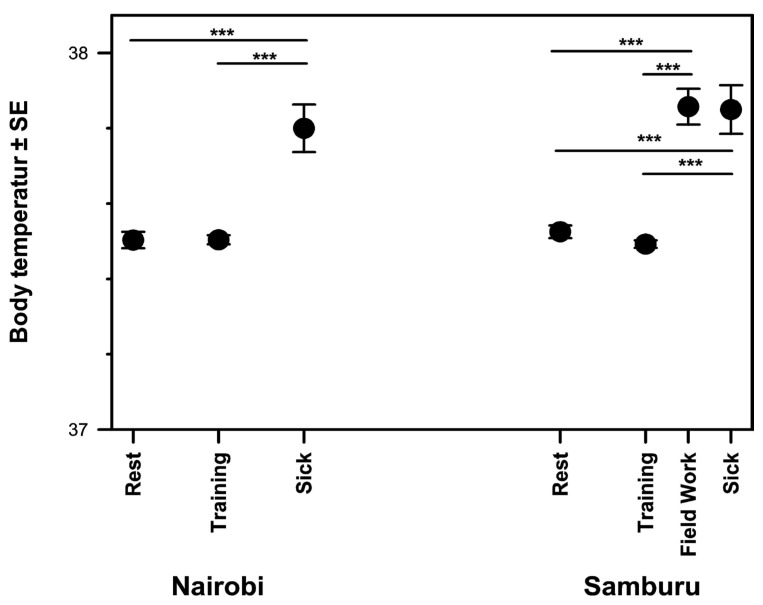
Mean body temperature of scat dogs one day after different activities. There was no difference between activities at different locations. After days of fieldwork, the body temperature increased, and there was no difference in the body temperature when a dog was sick (***, *p* < 0.001).

**Figure 3 animals-14-02456-f003:**
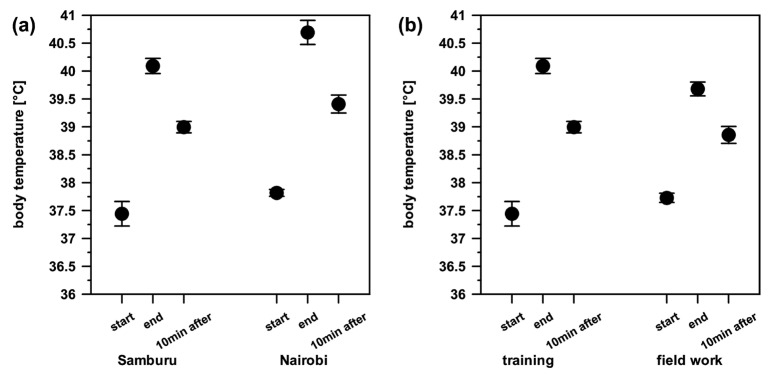
Change in body temperature ± S.E. after different activities and in different locations, immediately after exertion, and after 10 min of recovery. No difference was found between location (**a**) and activities (**b**) for body temperature after exertion and after recovery were detected.

**Figure 4 animals-14-02456-f004:**
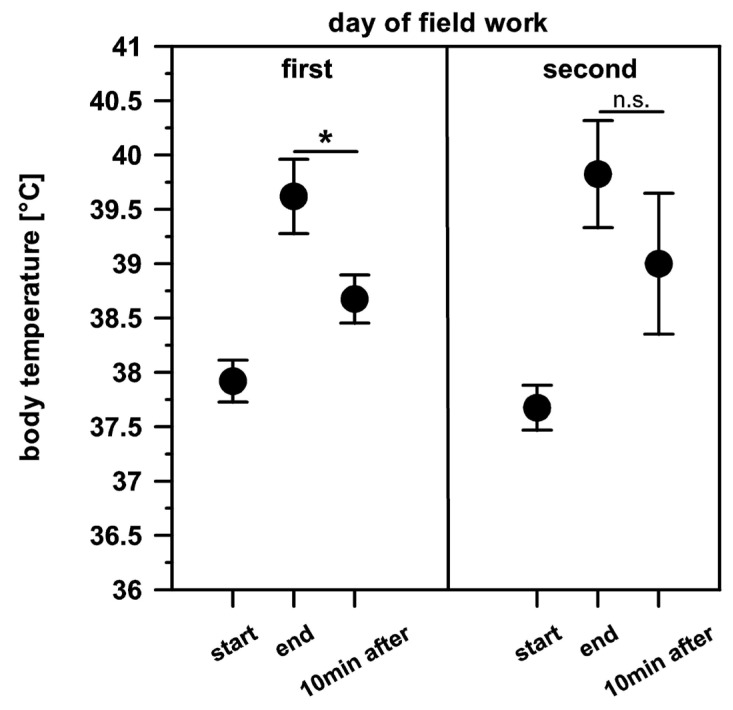
Change in body temperature ± S.E. after fieldwork of two dogs, immediately after exertion and after 10 min of recovery. On the first day, body temperature differs significantly directly after exertion and after 10 min of recovery (*, *p* < 0.05) but not on the second consecutive day of fieldwork (n.s., *p* > 0.05).

**Figure 5 animals-14-02456-f005:**
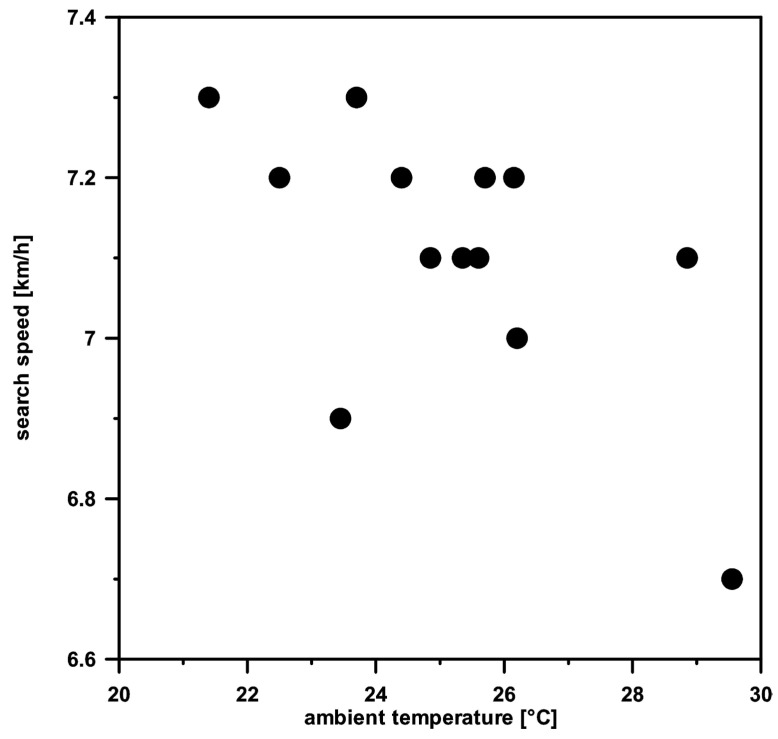
Relationship between ambient temperature and search speed for Madi only (Pearson correlation, r = −0.615, *p* < 0.05).

**Table 1 animals-14-02456-t001:** Characteristics of the dogs (name, sex, breed, coat color, and date of birth) included in the study, as well as the time range of available dates for each dog.

Name	Sex	Breed	Coat Color	Born	Data Available
Warrior	female	German Shepherd/Malinois	Brown	2014	1 April 2019–31 January 2021
Madi	male	Border Collie/Rottweiler	Black	March 2016	1 April 2019–27 February 2022
Arti	female	Malinois	Brown	April 2019	1 July 2019–31 December 2021
Persi	female	Malinois	Brown	April 2019	1 July 2019–27 February 2022

## Data Availability

The original contributions presented in the study are included in the Appendix A; further inquiries can be directed to the corresponding author.

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
