# Peer review of "Detection Dogs Working in Hot Climates: The Influence on Thermoregulation and Fecal Consistency"

_animals, 2024, doi:10.3390/ani14172456_

Round 1
Reviewer 1 Report
Comments and Suggestions for Authors
This an interesting study, which will be of interest to a wide range of practitioners and will be useful in improving welfare outcomes for detection dogs. I have made some specific comments in the attachment, but generally the manuscript is well written. The major comment I would make are that more information about the subject dogs should be included in table 1 (see attachment) and that the authors should include a justification for using stool consistency as a measure. I understand that stool consistency is largely a function of water content, but that should be explicitly stated in the methods and a reference cited.
The sample size is small but I understand that wildlife detection dogs are still relatively rare and finding other subjects would be difficult. This is also acknowledged by the authors.

Author Response
Thank you very much for your positive feedback and useful comments.
Since age, body mass, coat length and perhaps duration of working life (as a surrogate for physical fitness) may all affect the dog’s body temperature after exertion, those parameters should be included in this table.
We agree but we did not analyse the influence of these factors resp. the sample size of 4 makes it impossible. Therefore, we try to keep it simple and provide the reader with the most useful information only.
We adressed stool consitency as a measure in the methode section.
Reviewer 2 Report
Comments and Suggestions for Authors
PEER REVIEW
Detection Dogs Working in Hot Climates: The Influence on Thermoregulation and Body Condition
**please see PDF of paper with additional comments and suggested amended/expansions/clarifications**
Overall comments
Thank for you the opportunity to review this interesting and relevant paper exploring the impact of detection dog work on body temperature and stool quality.
This is a current area of interest and relevant in terms of performance, health and welfare for all dogs and there is a current, increasing body of work helping us to understand how best to support dogs and manage their thermoregulatory mechanisms – I will note that some very current references relating to this topic are not referenced and would suggest that the authors explore these as part of the paper editing and review - Access to our research | Hot Dogs – heatstroke education for dog owners
This is a research article and considers the background to study well, including setting the scene as to challenged faced by detection dogs. I would like more supporting evidence in many areas – highlighted on the script as this will add to the background and premise of the work. Indeed, I could suggest this is reconsidered as a descriptive case study more than a research article.
The sample size is very small and variable that is significant in the results presented – I feel there is insufficient critique of the variation within the study population and the likely impacts on collected data – see comments on the script please.
Study design and protocol appear robust and appropriate, but I feel there is information missing that is warranted in terms of materials and methods and relating to aspects of the work undertaken by the dogs including training duration and other differences in management/transport/handling etc that could impact on thermoregulatory and physiological responses.
· More study detail on dog management and demographics – diet etc (as this can have a significant impact on body temp)
· Nutritional detail is a little lacking – macronutrient profile? Feeding management?
· Ethical review??
· I feel that some outcomes are noted without real justification – perhaps more credence paid to observations and casual associations than is justified.
The premise of the work is clear, and the undertaking and analysis appears mostly appropriate but there are areas I feel warrant review and additional development within the document to support the conclusions (or these need tempered to highlight this is more of a case study than a robust research article)
Overall structure, flow and language use is mostly clear and supports readability but there are many typos that need addressing and some areas within figures where data appears missing. Comments are detailed within the manuscript for specific consideration.
Overall, I feel the work is robust and solid but does have substantial scope to further develop some areas of description and critical discussion, including avoidance of extreme extrapolation based on a small sample and limited experimental assessment.
Keywords – relevant and appropriate with exception of considering detection dogs instead of scat dogs to enhance searches highlighting this work
Figures and tables are fine and suitable for the work but some might benefit for critical review of data presentation (including missing data points?)
References
I have not exhaustively gone through these, but all appear fine – present and correct although I have not proofread or cross referenced to check validity of use.

Generally fine - some typos and word choice could be improved for clarity.
See comments on file
Author Response
Detection Dogs Working in Hot Climates: The Influence on Thermoregulation and Body Condition
Thank you very much for your details comments and the positive feedback
**please see PDF of paper with additional comments and suggested amended/expansions/clarifications**
We tried to present as much physical characteristics of the dogs as possible and necessary. Because of small sample size, it is hard to discuss the effect of these parameters.
Overall comments
Thank for you the opportunity to review this interesting and relevant paper exploring the impact of detection dog work on body temperature and stool quality.
This is a current area of interest and relevant in terms of performance, health and welfare for all dogs and there is a current, increasing body of work helping us to understand how best to support dogs and manage their thermoregulatory mechanisms – I will note that some very current references relating to this topic are not referenced and would suggest that the authors explore these as part of the paper editing and review - Access to our research | Hot Dogs – heatstroke education for dog owners
Thank you vey much for this information. We reviewed the suggested literatur and included it into the manuscript.
This is a research article and considers the background to study well, including setting the scene as to challenged faced by detection dogs. I would like more supporting evidence in many areas – highlighted on the script as this will add to the background and premise of the work. Indeed, I could suggest this is reconsidered as a descriptive case study more than a research article.
We agree with the conclusion and it is up to the editor to decide the type of manuscript.
The sample size is very small and variable that is significant in the results presented – I feel there is insufficient critique of the variation within the study population and the likely impacts on collected data – see comments on the script please.
Study design and protocol appear robust and appropriate, but I feel there is information missing that is warranted in terms of materials and methods and relating to aspects of the work undertaken by the dogs including training duration and other differences in management/transport/handling etc that could impact on thermoregulatory and physiological responses.
There was no influcence of training duration (line 208) and there are no differences in management/transport/handling because all dogs are managed in the same way by the same personal.
- More study detail on dog management and demographics – diet etc (as this can have a significant impact on body temp)
- Nutritional detail is a little lacking – macronutrient profile? Feeding management?
- Ethical review??
- I feel that some outcomes are noted without real justification – perhaps more credence paid to observations and casual associations than is justified.
The premise of the work is clear, and the undertaking and analysis appears mostly appropriate but there are areas I feel warrant review and additional development within the document to support the conclusions (or these need tempered to highlight this is more of a case study than a robust research article)
Overall structure, flow and language use is mostly clear and supports readability but there are many typos that need addressing and some areas within figures where data appears missing. Comments are detailed within the manuscript for specific consideration.
Overall, I feel the work is robust and solid but does have substantial scope to further develop some areas of description and critical discussion, including avoidance of extreme extrapolation based on a small sample and limited experimental assessment.
Keywords – relevant and appropriate with exception of considering detection dogs instead of scat dogs to enhance searches highlighting this work
Figures and tables are fine and suitable for the work but some might benefit for critical review of data presentation (including missing data points?)
References
I have not exhaustively gone through these, but all appear fine – present and correct although I have not proofread or cross referenced to check validity of use.
Softer stool and elevated body temperature, suggest this is a little too vague – for the simple summary we like to keep it simple
detect rather than identify? done
Perhaps add short detail as to why this is the case – we think, this would be to complicated for the summary.
Any details on how heat can impact performance? I am sure i have seen data on how scenting performance decreases as body temp increases in the detection dog literature
During panting the dogs are not able to sniff. There is no possibility to collect sent for further analysis
any details on ethical review and consideration please?
Ethical review was included at the end of the manuscript
very small sample that does need acknowledged as there is real variation in breed/type, sex, colour and likely acclimatisation
Was acknowledged in the discussion
no detail on other management aspects of dogs - housing breifly mentioned but diet? access to water? supplementation etc??
We agree, that all these factors may influence thermoregulatory abilities, but all the dogs were managed as a group and housing conditions, feeding and access to water were all the same and we did not analyse the influence of these factors. Therefore, we did not go into detail.
detection dog data from 3 month old pups? I feel this time period in relation to the age of the dog needs consideration as age will absolutely impact on ability to manage response to heat
We agree and considered it in the analysis
this feels VERY low? How typical was this? were thermometers calibrated and checked??
i am only talking from personal and research experience, but these temps do feel unusually low, certainly in comparison to dogs I have worked with - warrants consideration or at least acknowledgment in relation to other research about canine body temp
You are totally right, We got the wrong numbers
data points appear to be missing from figure? training at nairobi? travel values from both?
Figure was changed
perhaps detail on how training activity might differ from field work is warranted to help explore impact
This was described in the method section
Range is larger however and sample size is small - not sure how robust we can really say this data is but this is a fair comment. I would like to see in graphs/figures n = because it would be good to know if data is always from all four dogs or sometimes from fewer
We agree with the robustness of the data, but this is what we have and what the statistics tells us. With more dogs maybe, you find a significance during the second day too. Therefor we discussed the topic very carefully
I am genuinely not entirely sure if this figure in its presentation reveals anything significant and I feel that temp is almost the constant variable and should be x axis (forgive me if i have missed the point here!) I suspect other aspects are at play here too?
You are right and we changed the figure
vague comment and doesn´t take account of the other variables at play
We tried to explain the mechanism in principle, that’s the reason why we did not name values
I would disagree with this statement and note that for many working dogs, they will work through physiological discomfort (and many handlers miss it) so experience heat related illness and the consequences of such.
This is maybe true for agility or obedience dogs which are pushed by the handler. For search dog we know, that to don´t develop muscle soreness because they may regulate themselves. Maybe this is also true for regulating body temperature.
was there any data on water and food intake and any other physiological measures (urine output/faecal volume) that might be releavnt too? If not, perhaps worthy of discussion
After a working day nothing changed except no training in the afternoon. There are no other physiological measurements available.
this is not explicitly clear form your presented data? can you refer the reader to this data clearly?
This was described in chapter 3.4
yes, but has been noted in other working dogs as being 'normal' - work by John Houlton demonstrated this
We agree and find the same result. Still by definition it is in the range of fever
clarify CTM?
Was clarified in the introduction
actually, perhas not reduced activity but mofified activity - increased gut transit time and reduced absorption? caecal slap? issues with water intake and resorption? Reduced blood flow to and from GI tract? Lots of other GI factors potentially at play INCLUDING microbiome changes. These need noting and considering as variables and potentially causative factors
We changed the statement to signs of increased stress.
I would consider the validity of this dog being included in the study and this data not being previously highlighted as a significant variable likely to impact on data. I am also a little concerned about the ethical implications here of a study on a dog with a potential physiological compromise. However, heart murmurs can also appear in very fit animals too as an incidental finding, so might not be explicitly linked.
Immediately after the incident, the dog was retired and only the analysis of these data after years did show the difference on body temperature even when there were no symptoms before.
please do temper this with the fact that two of the four dogs started the study time period as immature puppies
done
do not disagree BUT lots of data now exisits about active dogs in differeing climates and there is real interest in this work so i feel this should not and would not be a limiting factor in the future, especially with the number of dogs deployed in this type of work.
This is true and maybe this publication is the impulse to put together all available data and do a more sophisticated analysis of the problem
and other physiological measures are restored too - panting alone will stop/reduce but physiology will still be in a lag of recovery.
Yes, off course. But panting you will see in the field and we think it not realistic to take other standard measurements during a search
i feel this needs tempered and noted that there is a suggestion that field work does impact on the ability of dogs to thermoregulatorily recover
Considered in the lower part of the paragraph
Reviewer 3 Report
Comments and Suggestions for Authors
This study present a very interesting data collection from detection dogs working in the field.
The association between physiological parameters and work performance is very important for handlers and researchers working with detection dogs, especially in extreme environments.
The introduction provides an exhaustive review of the state of art about the correlations between working dogs performances and physiological parameters, as well as the different methodologies and scores used.
The methods section nicely schematise the categorisation of the data collected and the rationale. However, some more precise information need to be provided such as models of the instruments used (thermometers) and descriptions of the scale used. In the statistical analyses as well it would be appreciated to have more details about the fixed effects included in the GLM.
Minor comments :
Line 9: “to coll down the body temperature”
Line 10: “we investigated the effect”
Line 11: “of dogs searching for scats in Kenya”
Comparison between body temperature in training days and resting days.
Line 26: Changing “search dogs” with “detection dogs”?
Which kind of thermometer did you use?
Results
I would suggest to present Estimates, z or t value and CONFIDENCE INTERVALS for the significant effects of the models presented.
Lines 226
How was the speed measured during the searches.
I would suggest to substitute the graphs with a boxplot indicating also the variation in the sample size.
Discussion
Did the authors assess the training efficiency of the dogs instead of just the speed of search?
Comments on the Quality of English LanguageThe quality of English is good and just need minor revisions.
Author Response
This study present a very interesting data collection from detection dogs working in the field.
The association between physiological parameters and work performance is very important for handlers and researchers working with detection dogs, especially in extreme environments.
The introduction provides an exhaustive review of the state of art about the correlations between working dogs performances and physiological parameters, as well as the different methodologies and scores used.
Thank you very much for your detailed and positive comments
The methods section nicely schematise the categorisation of the data collected and the rationale. However, some more precise information need to be provided such as models of the instruments used (thermometers) and descriptions of the scale used. In the statistical analyses as well it would be appreciated to have more details about the fixed effects included in the GLM.
Model of the thermometer was specified
As described in the analysis section, individual dog, activity and location were used as fixed effects. This is the same for temperatur before and after exertion.
Minor comments :
Line 9: “to coll down the body temperature”
Line 10: “we investigated the effect”
Line 11: “of dogs searching for scats in Kenya”
All done
Comparison between body temperature in training days and resting days.
Line 26: Changing “search dogs” with “detection dogs”?
Done
Which kind of thermometer did you use?
Done
Results
I would suggest to present Estimates, z or t value and CONFIDENCE INTERVALS for the significant effects of the models presented.
We additionally presented the test values
Lines 226
How was the speed measured during the searches.
See line 146 - search speed was calculated using the GPS collar of the dog
I would suggest to substitute the graphs with a boxplot indicating also the variation in the sample size.
We prefer to keep it simple and present mean ± SE. Makes the figure easier to read and understand
Discussion
Did the authors assess the training efficiency of the dogs instead of just the speed of search?
In this case we looked at two dogs only with the same amlunt of training. In our opinion main difference was the hear murmur of one dog. Because of small sample size it is difficult to discuss influence of body compostion, gender, health status and training efficiency.
Reviewer 4 Report
Comments and Suggestions for Authors
Overall:
Very good study regarding the health and safety of search dogs in hot environments. See additional notes for suggested improvements.
Simple Summary:
Line 10: change “environment” to “environments”
The Simple Summary should include some additional details regarding how dogs typically cool themselves, the climate in Kenya that might pose a problem to the dogs, and why this combination is of particular concern to detection/search dogs in Kenya
Abstract:
The abstract is good overall, but needs to include the number of dogs in the experiment.
Introduction:
Overall very good overview of the background around dog physiology and health with relation to heat and homeostasis.
Lines 50-53: Include a citation for the sweat glands in their paws and the other factors contributing to dissipation of heat.
Materials and Methods:
Overall very thorough and included the appropriate amount of detail.
Line 88: With regards to the “main base”, authors should specify if this is the main base for Kenya or other smaller areas.
Line 125-126: Include a citation for the body condition score for dogs.
Results:
Overall, the authors used appropriate statistics to analyze the data and the results were thorough and easy to understand.
Discussion and Conclusion:
Overall, the discussion and conclusion sections provide sufficient context for the findings of the study and adequately address the limited sample size and geographic location of the experiment. It would be beneficial for the authors to include suggestions of how the study might be replicated in other environments to control for the climate.
Comments on the Quality of English Language
There are minor spelling/grammar issues that can be easily corrected.
Author Response
Overall:
Very good study regarding the health and safety of search dogs in hot environments. See additional notes for suggested improvements.
Thank you very much for your comments and the positive feedback.
Simple Summary:
Line 10: change “environment” to “environments” - done
The Simple Summary should include some additional details regarding how dogs typically cool themselves, the climate in Kenya that might pose a problem to the dogs, and why this combination is of particular concern to detection/search dogs in Kenya
We added the suggested information
Abstract:
The abstract is good overall, but needs to include the number of dogs in the experiment.
We added the suggested information
Introduction:
Overall very good overview of the background around dog physiology and health with relation to heat and homeostasis.
Lines 50-53: Include a citation for the sweat glands in their paws and the other factors contributing to dissipation of heat.
Was added
Materials and Methods:
Overall very thorough and included the appropriate amount of detail.
Line 88: With regards to the “main base”, authors should specify if this is the main base for Kenya or other smaller areas.
It´s the organizations main base
Line 125-126: Include a citation for the body condition score for dogs.
Was added
Results:
Overall, the authors used appropriate statistics to analyze the data and the results were thorough and easy to understand.
Discussion and Conclusion:
Overall, the discussion and conclusion sections provide sufficient context for the findings of the study and adequately address the limited sample size and geographic location of the experiment. It would be beneficial for the authors to include suggestions of how the study might be replicated in other environments to control for the climate.
We refer to Schneider, M.; Slotta-Bachmayr, L., Physical and Mental Stress of SAR Dogs during Search Work. In Canine Ergonomics: The Science of Working Dogs, First ed.; Helton, W. S., Ed. CRC Press: 2009; pp 263-279. where similar data have been collected at least befor and after extertion. But we added a suggestion to replication in different environments
Round 2
Reviewer 2 Report
Comments and Suggestions for Authors
thank you for making some revisions and edits based on review - i hope they were useful and that you feel they have enhanced the work.
Only a couple of minor amends suggested - please see attached file with comments and highlighted areas.

Author Response
Thank you very much for your comments. They were very useful and helped a lot. All of your remarks were considered.